# Serum Metabolome Adaptations Following 12 Weeks of High-Intensity Interval Training or Moderate-Intensity Continuous Training in Obese Older Adults

**DOI:** 10.3390/metabo13020198

**Published:** 2023-01-29

**Authors:** Layale Youssef, Mélanie Bourgin, Sylvère Durand, Fanny Aprahamian, Deborah Lefevre, Maria Chiara Maiuri, Vincent Marcangeli, Maude Dulac, Guy Hajj-Boutros, Fanny Buckinx, Eva Peyrusqué, Pierrette Gaudreau, José A. Morais, Gilles Gouspillou, Guido Kroemer, Mylène Aubertin-Leheudre, Philippe Noirez

**Affiliations:** 1School of Kinesiology and Physical Activity Sciences, Faculty of Medicine, Montreal, QC H3T 1J4, Canada; 2Research Center of the Montreal Geriatrics Institute (CRIUGM), Montreal, QC H3W 1W5, Canada; 3INSERM U1138, Centre de Recherche des Cordeliers, Sorbonne Université, Université Paris Cité, 75006 Paris, France; 4Metabolomics and Cell Biology Platforms, Gustave Roussy Cancer Campus, AMMICa US23/CNRS UMS3655, 94805 Villejuif, France; 5Département des Sciences Biologiques, Faculté des Sciences, Université du Québec à Montréal, Montreal, QC H2X 1Y4, Canada; 6Département des Sciences de l’Activité Physique, Faculté des Sciences, Université du Québec à Montréal, Montreal, QC H2X 1Y4, Canada; 7Department of Medicine, Research Institute of the McGill University Health Centre, Montreal, QC H4A 3J1, Canada; 8Groupe de Recherche en Activité Physique Adaptée, Montreal, QC H2X 1Y4, Canada; 9Département de Médecine de l’Université de Montréal, Centre de Recherche du Centre Hospitalier de l’Université de Montréal (CRCHUM), Montreal, QC H2X 3E4, Canada; 10Performance, Santé, Métrologie, Société (PSMS), UFR STAPS, Université de Reims Champagne Ardenne, 51100 Reims, France; 11T3S, Inserm, Université Paris Cité, 75006 Paris, France; 12Institut de Recherche Médicale et d’Épidémiologie du Sport (IRMES), Institut National du Sport, de l’Expertise et de la Performance (INSEP), Université Paris Cité, 75012 Paris, France

**Keywords:** metabolomics, mass spectrometry, aging, obesity, HIIT, MICT

## Abstract

Physical activity can be effective in preventing some of the adverse effects of aging on health. High-intensity interval training (HIIT) and moderate-intensity continuous training (MICT) are beneficial interventions for the quality of life of obese older individuals. The understanding of all possible metabolic mechanisms underlying these beneficial changes has not yet been established. The aim of this study was to analyze changes in the serum metabolome after 12 weeks of HIIT and MICT in obese older adults. Thirty-eight participants performed either HIIT (*n* = 26) or MICT (*n* = 12) three times per week for 12 weeks. Serum metabolites as well as clinical and biological parameters were assessed before and after the 12-week intervention. Among the 364 metabolites and ratio of metabolites identified, 51 metabolites changed significantly following the 12-week intervention. Out of them, 21 significantly changed following HIIT intervention and 18 significantly changed following MICT. Associations with clinical and biological adaptations revealed that changes in acyl-alkyl-phosphatidylcholine (PCae) (22:1) correlated positively with changes in handgrip strength in the HIIT group (r = 0.52, *p* < 0.01). A negative correlation was also observed between 2-oxoglutaric acid and HOMA-IR (r = −0.44, *p* < 0.01) when considering both groups together (HIIT and MICT). This metabolite also correlated positively with quantitative insulin-sensitivity check index (QUICKI) in both groups together (r = 0.46, *p* < 0.01) and the HIIT group (r = 0.51, *p* < 0.01). Additionally, in the MICT group, fumaric acid was positively correlated with triglyceride levels (r = 0.73, *p* < 0.01) and acetylcarnitine correlated positively with low-density lipoprotein (LDL) cholesterol (r = 0.81, *p* < 0.01). These four metabolites might represent potential metabolites of interest concerning muscle strength, glycemic parameters, as well as lipid profile parameters, and hence, for a potential healthy aging. Future studies are needed to confirm the association between these metabolites and a healthy aging.

## 1. Introduction

Exercise, well known for its major health benefits, is considered an effective non-pharmacological strategy for the elderly [1]. Obesity, whose prevalence increases with aging [2], has also become a major public health issue. Thus, when low muscle mass and function are combined with excess adiposity, being obese and old becomes a double health burden [3]. In terms of clinical and biological adaptations, high-intensity interval training (HIIT) and moderate-intensity continuous training (MICT) are effective interventions to improve the health status of obese older individuals [4,5]. More specifically, it was previously demonstrated that MICT decreased relative gynoid fat mass and increased lower limb muscle strength, whereas HIIT increased functional capacities and lean mass in obese older adults [4]. Since the impacts of HIIT and MICT on serum metabolites in obese older adults remain largely unexplored, it becomes highly interesting to evaluate the possible mechanisms underlying the changes following these interventions.

With advances in mass spectrometry, the number of metabolomics studies has increased. The metabolomic approach has recently become an area of interest in the field of exercise science [6,7]. Physical activity interventions can induce metabolic adaptations through changes in serum metabolomic profile [8]. However, there is still little information regarding changes after different types of exercise interventions and in different types of populations.

With respect to mode of exercise, it has been shown that changes in the plasma metabolome can be both mode-dependent and mode-independent, meaning that the molecular response must be interpreted in the context of the mode by which the biological changes occur [9]. In participants with an average age of 62 years, enrichment of pathways involving connective tissue metabolism and lipid signaling was found in plasma samples after treadmill exercise (3 times per week for 50 min), whereas branched-chain amino acids (BCAA), tryptophan, tyrosine, and urea cycle were altered after stretching (twice per week for 50 min) [10]. One study examined the effect of HIIT and MICT on metabolism and counterregulatory stress hormones in well-trained male cyclists and triathletes. The results revealed that carbohydrate oxidation was higher, and fat oxidation was lower following HIIT [11]. Furthermore, metabolomic analysis revealed that tricarboxylic acid (TCA) intermediates and monounsaturated fatty acids increased after HIIT and BCAA decreased after both HIIT and MICT [11]. Therefore, different adaptations were observed following different training modalities.

In addition, metabolic adaptations are not the same for all types of populations (age and health status). In healthy young soccer players, analysis of urine sample metabolites was performed before and after a HIIT intervention. Following a 2-day running HIIT session, metabolomic changes included downregulation of steroid hormone metabolites and upregulation during recovery, suggesting increased muscle growth after HIIT [12]. In middle-aged obese men and women, after an exercise intervention (walking/jogging on a treadmill at multiple intensities 5 times per week for 24 weeks), several changing metabolites, such as TCA cycle intermediate (isocitric acid), BCAA, gluconeogenic amino acids, as well as xanthurenic acid, were associated with changes in cardiometabolic risk traits [13]. In a study of participants (aged 30–60 years) with higher risk factors for metabolic syndrome and greater adiposity, serine and glycine were found in lower concentrations. However, when activity measures (measured by estimated calculations) were increasing, their concentrations were higher [14]. Additionally, in patients with pre-diabetes and type 2 diabetes, HIIT was found to be a more time-efficient strategy than MICT as serum ficolin-3 levels (associated with diabetes) decreased after 3 weeks of HIIT training [15]. In addition, in sedentary overweight and obese women (with an average age of 27 years), changes in glucose tolerance were predicted using personalized metabolomics after a 6-week HIIT intervention (cycling for 30 min twice a week) [16]. To our knowledge, no study has explored adaptations in the serum metabolome after HIIT and MICT in obese older men and women. Therefore, the objective of this study was to analyze changes in serum metabolites after 12 weeks of HIIT or MICT in obese older adults.

## 2. Materials and Methods

### 2.1. Study Design

This study, which combined two randomized controlled trials, is a sub-analysis of our previous studies [4,5]. The ethics committee of the “Université du Québec à Montréal (UQAM)” approved all procedures (#2014_e_1018_475). After being informed about the study’s purpose, aim, procedures and associated risks, the participants provided their informed written consent.

### 2.2. Participants

Participants were recruited from the community via social communication (flyers and meetings in community centers) in the Greater Montreal area. The list of inclusion and exclusion criteria, which was previously described [4,5], is detailed in the Appendix A.

To be considered as having completed the intervention, participants had to complete at least 80% of the training sessions (minimum: 29/36 sessions). Among the participants included in the previous analysis [4], only those with pre- and post-intervention serum metabolites analyses (HIIT: *n* = 26 vs. MICT: *n* = 12) were included in this study.

### 2.3. Exercise Interventions

All the participants performed 3 supervised training sessions per week (HIIT or MICT) during 12 consecutive weeks.

#### 2.3.1. High-Intensity Interval Training (HIIT)

Under the supervision of trained kinesiologists (i.e., certified exercise instructors), participants performed the HIIT training on an elliptical device (TechnoGym Synchro Exc 700, Technogym, NJ, USA ) to reduce impacts. The HIIT session lasted 30 min and was divided as follows: (1) five-minute warm-up at a low intensity (50–60% maximal heart rate (MHR) and/or 8–12 on Borg’ scale); (2) twenty minutes of HIIT consisting of multiple 30-s high-intensity sprints (80–85% MHR or >17 on Borg’ scale) alternated with 90 s at moderate intensity (65% MHR or 13–16 on Borg’ scale); and (3) five-minute cool down period (50–60% MHR or 8–12 on Borg’ scale). To determine the intensity of each cycle, MHR percentage and/or perceived exertion (Borg scale; relying exclusively on perceived exertion for participants using anti-arrhythmic and inotropic agents) were used. The MHR was determined using the following equation: [((220 − age) − Heart Rate rest) × % Heart Rate target] + Heart Rate rest. Continuous adjustment of treadmill speed and resistance during the HIIT intervention ensured that MHR was always above 80% during high intensity intervals.

#### 2.3.2. Moderate-Intensity Continuous Training (MICT)

Under the supervision of trained kinesiologists (i.e., certified exercise instructors), participants underwent MICT by walking on a treadmill (Precor C936i, Precor, WA, USA). The session lasted 1 h and the MICT was performed at a moderate intensity (60–70% MHR or 13–14 on the Borg scale). Continuous adjustment of treadmill speed and resistance ensured that MHR was maintained between 60–70% MHR or 13–14 on the Borg scale.

### 2.4. Clinical Parameters

The detailed assessment of the clinical parameters was previously described in Youssef et al., 2022 [4]. To evaluate physical performance and muscle function, the validated tests used were previously described in Buckinx et al., 2018 [17].

The five tests used to assess the physical performance were the 6-min walk test [18,19], the walking speed [20,21], the unipodal balance [22], the timed up and go [23], the chair stand test [24] and the step test [25]. Three tests were used to assess muscle function: grip strength, lower limb muscle power, and lower muscle strength. These measures of muscle function were expressed in absolute units (kg or W or N, respectively) and normalized to body weight and lean limb mass. To evaluate anthropometric characteristics, body weight (kg) and height (m) were determined, which allowed calculation of the BMI (body mass (kg)/height (m^2^)).

To assess the body composition, fat masses (total, android, gynoid, arm and leg; %) and total lean masses (total, arm and leg; kg) were quantified in fasted state.

Muscle area, subcutaneous, and intramuscular fat content were used to assess thigh composition.

### 2.5. Biological Parameters

Fifteen milliliters of blood were collected from each participant to assess fasting serum levels of biochemical and hormonal markers following a 12-h overnight fast. The detailed assessment of the biological parameters was previously described in Marcangeli et al., 2022 [5]. Briefly, the lipid profile was assessed through total, HDL- and LDL-cholesterol, as well as triglyceride levels. Adipose tissue metabolites and adipokines (free fatty acids, adiponectin and leptin levels, adiponectin/leptin ratio), members of the insulin-like growth factor (IGF) family (IGF1; IGFBP3 and IGFB3/IGF1 molar ratio) and glucose-insulin homeostasis (glucose and insulin levels but also HOMA and QUICKI indices) were assessed.

### 2.6. Metabolomic Profiling

Blood samples were collected by a physician in the fasting state before and after the 12-week intervention. Serum samples were used to blindly analyze serum metabolites at the Gustave Roussy Cancer Campus facility (Villejuif, France) using mass spectrometers coupled to multiple different liquid or gas phase chromatography methods. Bile acids metabolomics were obtained using a UHPLC/MS—RRLC 1260 system (Agilent Technologies, Waldbronn, Germany) coupled to a Triple Quadrupole 6410 (Agilent Technologies). Short chain fatty acids, oxylipin, and lipids metabolomics were assessed by a UHPLC/QUAD+—RRLC 1260 system (Agilent Technologies, Waldbronn, Germany) coupled to a 6500+ QTRAP (Sciex, Darmstadt, Germany). Polyamines metabolomics were quantified using a UHPLC/QQQ—RRLC 1260 system (Agilent Technologies, Waldbronn, Germany) coupled to a Triple Quadrupole 6410 (Agilent Technologies). Concerning the level of identification, for the targeted metabolomics it is identified (level 1, validated by standards injections on the Orbitrap, and multiple reaction monitoring MRM developments for the LCQQQ and GCQQQ), and for the metabolomic profiling it is putative. Intra batch correction was performed based on quality control pool and processed on R software. The details of each method were previously described [26,27].

### 2.7. Statistical Analyses

Quantitative results are expressed as mean ± SD. The delta changes (%) were calculated as (post − pre)/pre × 100. Statistical significance tests of the measured metabolites were performed using multivariate and univariate analyses. For the multivariate analysis, principal component analysis (PCA) was conducted to reduce the dimensionality of the data (R-packages FactoMineR and factoextra) and volcano plots were performed for data visualization (R-package tidyverse and ggrepe1). The Levene’s test was used to assess the homogeneity of variances. For the univariate analysis, a linear mixed-models approach (R-package nlme) with a two-factor repeated measures ANOVA was then used to test the intervention effect (Time effect), and the interaction effect (Time × Training effect) on serum metabolites adaptations. Post-hoc analyses were then done using simultaneous tests for general linear hypotheses (R-package emmeans) with a Bonferroni correction. Results were considered statistically significant when *p*-value < 0.05. A large number of comparisons were made, so the Benjamini Hochberg (BH) false discovery rate (FDR) was performed to correct for multiple comparisons (R-package FSA). The threshold for the FDR was set at 0.05. To assess the association between delta changes of the metabolites and the delta changes of the clinical and biological parameters, Pearson’s correlation coefficient analysis was then performed in the HIIT group. However, as the number of subjects was low in the MICT group, Spearman’s correlation coefficient analysis was used. Only correlations with Pearson’s and Spearman’s correlation coefficient above 0.35 and with *p*-value < 0.01 were considered. All statistical analyses were performed using the software R (3.6.2) (foundation for statistical computing, Vienna, Austria). Heatmaps of the metabolites were generated with the R-packages ggplot2 and tidyr, and the correlation graphs were drawn using the R-package ggpubr.

## 3. Results

### 3.1. Clinical and Biological Characteristics

The characteristics of the participants at baseline are present in Table 1. The clinical and biological characteristics of these participants were previously described [4]. However, for this sub-analysis, only those with pre- and post-intervention serum metabolites analyses were included (Appendix A). The significance of the clinical and biological parameters was almost similar to the previous study. However, due to the lower number of participants, the total lean mass and the gynoid fat mass, which appeared significant in the previous study, were not significant in the current sub-analysis (Appendix A).

### 3.2. Overall Metabolomic Profile Adaptations Following the 12-Week Intervention

Following the 12-week HIIT or MICT intervention, 364 serum metabolites and t ratio of metabolites were identified and quantified. The PCA analysis, which was performed following the overall 12-week intervention (Figure 1), revealed no significant changes.

### 3.3. Alterations of Energetic Metabolisms Following the 12-Week HIIT and MICT Intervention

To identify the best metabolites of interest, we generated a two-way repeated measures ANOVA. This analysis revealed significant changes (*p* < 0.05) for fifty-one metabolites involved in different metabolic pathways (Table 2 and Table 3): six belong to the tricarboxylic acid (TCA) cycle, five to the carbohydrate metabolism, eleven to the amino acid metabolism, and twenty-nine to the fat metabolism. Considering significant changes for each type of intervention alone, 21 metabolites significantly varied after the 12-week HIIT intervention (Figure 2A; abundance was normalized between 0 and 1), and 18 metabolites significantly varied after the 12-week MICT intervention (Figure 2B; abundance was normalized between 0 and 1).

For the TCA cycle (Table 2), following the 12-week intervention, five metabolites significantly changed following the HIIT intervention, and one metabolite significantly changed following the MICT intervention. More specifically, following HIIT, two metabolites increased (2-oxoglutaric acid and fumaric acid) while the three others (3-methylhistidine, aspartic acid and the ratio aspartate/malate) significantly decreased. Following MICT, 2-oxoglutaric acid significantly increased.

For the carbohydrate metabolism (Table 2), following the 12-week intervention, two metabolites significantly changed following the HIIT intervention, and two metabolites changed following the MICT intervention. More specifically, acetic acid and glyceric acid significantly decreased after HIIT, whereas xylitol and xylose significantly increased after MICT.

For the amino acid metabolism (Table 2), following the 12-week intervention, four metabolites significantly changed following the HIIT intervention, and five metabolites significantly changed following the MICT intervention. More specifically, three metabolites decreased (2 hydroxybutyric acid, 2-oxovaleric acid and inosine) while ketoisovaleric acid increased following HIIT. On the other side, the five metabolites that significantly changed after MICT (2-aminoadipic acid, hypotaurine, ornithine, uric acid and xanthine) significantly increased.

For the fat metabolism (Table 3), 10 metabolites significantly changed following the HIIT intervention, and 10 metabolites significantly changed following the MICT intervention. More specifically, following HIIT, two metabolites increased (Carnitine C18:0 and margaric acid), whereas eight metabolites significantly decreased (acetylcarnitine, Ceramide (C18:1/24:0), Diglyceride (DG) (18:1/18:3), DG (20:4/18:2), isobutyric acid, linoleic acid, acyl-alkyl-phosphatidylcholine (PCae) (16:0) and PCae (22:1)). Following MICT, an increase was observed for eight metabolites (DG(18:1/18:3), margaric acid, panthothenic acid, PCae(15:0), triglyceride (TG) (16:1/18:1/18:0), TG (16:1/18:3/18:2), TG (16:2/18:2/18:2) and undecanoic acid) and a decrease was observed for two metabolites (ether-phosphatidylethanolamine (PEee) (19:1) and TG (12:0/14:0/16:0)).

To strengthen our feature selection criteria, we undertook a method-based false discovery rate (FDR)-adjusted (Appendix A). Following this analysis, two metabolites revealed significant differences due to the 12-week intervention (Time effect), one metabolite due to HIIT effect, and eight metabolites due to MICT effect. More specifically, when the general intervention was considered, significant changes (FDR < 0.05) were observed for only two metabolites, which are 2-oxoglutaric acid and glyceric acid. For the HIIT group alone, only glyceric acid significantly changed (FDR < 0.01). For the MICT group alone, a significant change was revealed for eight metabolites (FDR < 0.05): 2-oxoglutaric acid, xylitol, ketoisovaleric acid, uric acid, DG (18:3/18:3), pantothenic acid, TG (16:1/18:3/18:2) and TG (16:2/18:2/18:2).

To identify the differences in metabolite abundance before and after the intervention, volcano plots were drawn (Figure 3). This shows that four metabolites were upregulated before MICT compared to five metabolites that were upregulated after MICT (Figure 3A). Before HIIT, one metabolite was upregulated compared to two metabolites that were upregulated after HIIT (Figure 3B). Additionally, twelve metabolites were upregulated after MICT compared to five metabolites that were upregulated after HIIT (Figure 3C).

### 3.4. Associations between Changes in Serum Metabolites and Changes in Clinical and Biological Parameters

Correlations between serum metabolites delta changes (%) and clinico-biological parameters delta changes (%) were obtained. Significant correlations (*p* < 0.01) resulting from the overall intervention, the HIIT intervention and the MICT intervention were obtained (Table 4, Table 5 and Table 6). The impacts of HIIT and MICT on the metabolites that were among the interesting correlations are presented in Figure 4. 

The delta change in the metabolite PCae (22:1) significantly correlated with the delta change in handgrip strength, a parameter of muscular strength, and an important clinical parameter in the elderly.

Following the FDR analysis, the delta change in PCae (22:1) did not appear significant. However, this metabolite was considered since interesting correlations were observed between PCae (22:1) and muscle strength parameters in the HIIT group (Table 4). More specifically, delta change of PCae (22:1) significantly correlated with delta changes of the handgrip strength (r_p_ = 0.52, *p* < 0.01; Figure 5A), the handgrip strength relative to body weight (r_p_ = 0.54, *p* < 0.01), and the handgrip strength relative to arms lean mass (r_p_ = 0.54, *p* < 0.01; Figure 5B). The delta changes of the three metabolites 2-oxoglutaric acid, fumaric acid and acetylcarnitine were significantly correlated with biological parameters delta changes (Table 6).

The change in 2-oxoglutaric acid appeared significant following the FDR analysis (FDR < 0.05). When the entire group of participants was considered, this metabolite delta changed significantly, correlated with delta changes of both HOMA-IR (r_p_ = −0.44, *p* < 0.01; Figure 5C) and QUICKI (r_p_ = 0.46, *p* < 0.01; Figure 5D), which are insulin sensitivity parameters [28]. Interestingly, 2-oxoglutaric acid delta change also correlated with QUICKI delta change (r_p_ = 0.51, *p* < 0.01; Figure 5E) in the HIIT group alone. The changes of both fumaric acid and acetylcarnitine did not appear significant following the FDR analysis, however interesting correlations with lipid profile parameters delta changes were observed in the MICT group. More specifically, fumaric acid delta change correlated well with triglycerides level delta change (r_s_ = 0.73, *p* < 0.01; Figure 5F), and acetylcarnitine delta change correlated well with LDL cholesterol (r_s_ = 0.81, *p* < 0.01; Figure 5G).

## 4. Discussion

### 4.1. Metabolome Adaptations

The metabolomic analysis was conducted on a population of obese older men and women following a 12-week HIIT or MICT intervention. Our aim was to identify possible serum metabolites of interest underlying the adaptations in clinical and biological parameters in obese older adults. Given that these HIIT and MICT modalities in this type of population have previously been shown to induce changes in clinical and biological parameters [4,5,17], metabolites changes could then be considered as molecular signatures of clinico-biological adaptations. We observed that several metabolites increased or decreased significantly as a result of the overall 12-week intervention. In addition, some of these metabolites were altered after HIIT or MICT, meaning that the metabolome undergoes specific changes after each type of exercise. Moreover, the metabolite DG (18:1/18:3) significantly decreased after HIIT but significantly increased after MICT. This counter change could allow us to speculate that the different metabolisms may not only be altered after exercise but may also be altered differently depending on different types of exercise. A study performed on athletes after a single bout of exercise showed that the serum metabolome of endurance and bodybuilding athletes was affected differently (28). This result suggests that the serum metabolome may be altered differently depending on the different types of exercise usually performed by the individual, as well as on different specific factors (athletic or non-athletic individuals). The fact that the serum metabolome underwent different adaptations is in line with our results, as the metabolome of our participants was also affected differently after different types of exercise. Interesting correlations between significantly modified metabolites and several clinico-biological parameters were obtained. To the best of our knowledge, this is the first study in which an analysis of the serum metabolites was performed following a 12-week HIIT or MICT intervention in obese older adults.

Our intervention consisted of two different exercise modalities with different intensities, and some metabolites changes were elicited by HIIT or by MICT. Interestingly, it was previously found that changes in the metabolome occurred following non-pharmacological interventions. Among these interventions, exercise [29] or nutrition [30] were able to induce changes in the metabolome. In a study evaluating the metabolome changes in adult men following a 3-h marathon [31], glyceric acid increased. In our study, a chronic intervention in older adults, this metabolite decreased following HIIT. This discrepancy could be caused by several factors, like the different age of the participants and the different frequency of intervention, with the marathon being an acute intervention and the HIIT being a chronic intervention over 12 weeks. Therefore, the metabolome is able to encounter different changes depending on different interventions and different factors (age, sex, and type of intervention modality).

### 4.2. PCae (22:1) as a Possible Metabolite of Interest for Muscle Strength

One of the major adverse health effects of aging is the loss of muscle strength. This loss becomes much faster than the loss of muscle mass in older individuals, resulting in a decline in muscle quality [32]. In addition, age and body fat have been shown to be inversely associated with muscle strength and quality [33]. Therefore, since obese individuals will experience an even faster decrease in muscle strength and quality, preventing fat gain and preserving lean mass becomes an important issue. As a standard clinical measure, it has been proposed that handgrip strength may be a marker of frailty assessment [34]. Following the 12-week HIIT intervention, a significant positive correlation was observed between the metabolite PCae (22:1) delta change and the handgrip strength delta change of our obese elderly participants. This metabolite is included in the category of fat metabolism metabolites. In a previous study aiming to identify metabolites of interest associated to handgrip strength decline in old participants (aged 55 years and above), it was proposed that acetylcarnitine might reflect specific perturbations occurring in the mitochondrial fat metabolism during aging [35]. In our sub-analysis, we did not find any significant correlation between acetylcarnitine and handgrip strength, which could be due first to the obesity status of our participants and second to the exercise intervention. However, our result could be partially in line with that of Siang Ng and colleagues [35], since the metabolite which correlated well with handgrip strength is also part of fat metabolism. More specifically, PCae is an abundant phospholipid in the cell membranes and plays a role in the regulation of the lipid metabolism [36]. Therefore, fat metabolism could play a major role in the decline of handgrip strength, and metabolites of this metabolism could be of interest for muscular strength.

### 4.3. Possible Metabolites of Interest for Glycemic and Lipid Profile Parameters in Obese Older Adults

In a study evaluating metabolic and hormonal responses to HIIT and MICT in well-trained male cyclists and triathletes, distinct changes in specific metabolites were found, suggesting that differences in metabolic demand affect the serum metabolome [11]. Interestingly, similarly to our results, they found that the majority of the TCA cycle intermediates increased after HIIT. Following our 12-week intervention, the metabolite 2-oxoglutaric acid, a TCA cycle intermediate, correlated well with HOMA-IR and QUICKI, two parameters of the glycemic control. Additionally, this metabolite correlated well with QUICKI in the HIIT group. Peake and colleagues did not report any significant change for this metabolite [11], and this discrepancy could be due to different factors such as different physical activity levels between the two cohorts of participants, obesity status and age. In the MICT group, interesting correlations were observed between delta changes of fumaric acid and triglycerides level as well as between delta changes of acetylcarnitine and LDL cholesterol, which are important lipid profile parameters. However, further research is needed to confirm these associations with a larger number of obese older participants undergoing MICT.

### 4.4. Limitations and Future Perspectives

This study presents some limitations. The number of participants between the two groups was not high (26 for HIIT and 12 for MICT) since not all participants from the a priori study had extra blood samples. Concerning the participants characteristics, our participants were moderately obese, and this might not reflect the more severely obese population. As only volunteer subjects were included in the 12-week intervention, a selection bias is also possible. Most importantly, different devices were used for the exercise intervention (elliptical trainer for HIIT and treadmill for MICT). As our participants might be osteoporotic, and HIIT might cause high surface impacts, an elliptical trainer was used to prevent joint injuries on the lower limbs. Hence, adding one group performing MICT on the elliptical trainer might be of interest in the future studies. For the metabolomic analyses, intermediate metabolomic analyses (for example in the middle of the 12-week intervention) are missing. Additionally, whole-body metabolisms cannot be represented by the serum metabolome, therefore additional analyses from different samples like plasma, urine, or saliva would be of interest. A replication of our results in an independent validation cohort undergoing the similar 12-week intervention is required. Our results were presented both with and without FDR analysis, to avoid missing any metabolite or ratio of metabolites that could present a potential interest. We tried to identify new putative metabolites of interest specific to HIIT and MICT in obese older adults that could be used to treat obesity and the decline in age-related functions. However, future studies are needed to confirm our results.

## 5. Conclusions

To the best of our knowledge, our sub-analysis study is the first to compare the adaptations in the serum metabolites following a 12-week HIIT or MICT in obese older adults. Our results showed that 51 metabolites from different biochemical pathways were affected by the 12-week intervention. Several of these metabolites correlated well with clinical and biological adaptations. Interesting associations were revealed between PCae (22:1) and handgrip strength, as well as between 2-oxoglutaric acid and glycemic parameters (HOMA-IR and QUICKI). Additionally, fumaric acid and acetylcarnitine correlated well with lipid profile parameters (triglycerides level and LDL cholesterol respectively). These metabolites were identified as possible metabolites of interest for muscle strength, glycemic parameters, and lipid profile parameters, and hence, for a potential healthy ageing.

## Figures and Tables

**Figure 1 metabolites-13-00198-f001:**
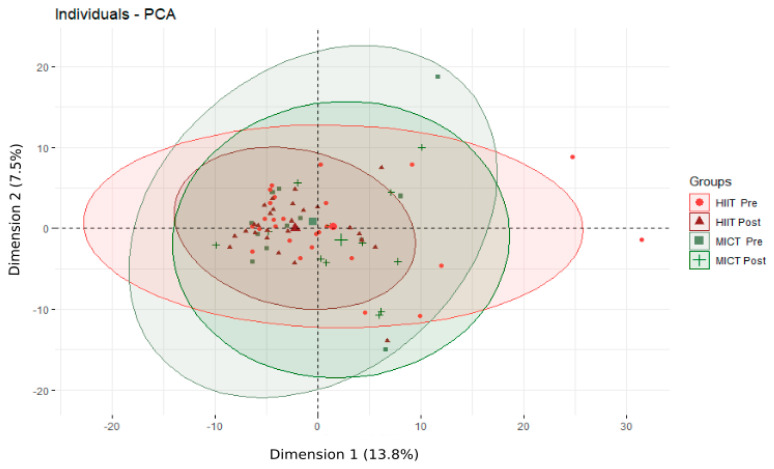
Principal Component Analysis (PCA) of the metabolome of obese older adults before and after the 12-week intervention. HIIT: high-intensity interval training; MICT: moderate-intensity continuous training; Pre: before the 12-week intervention; Post after the 12-week intervention; Individuals refer to participants; The % refers to the extent of variance reflected in the whole data.

**Figure 2 metabolites-13-00198-f002:**
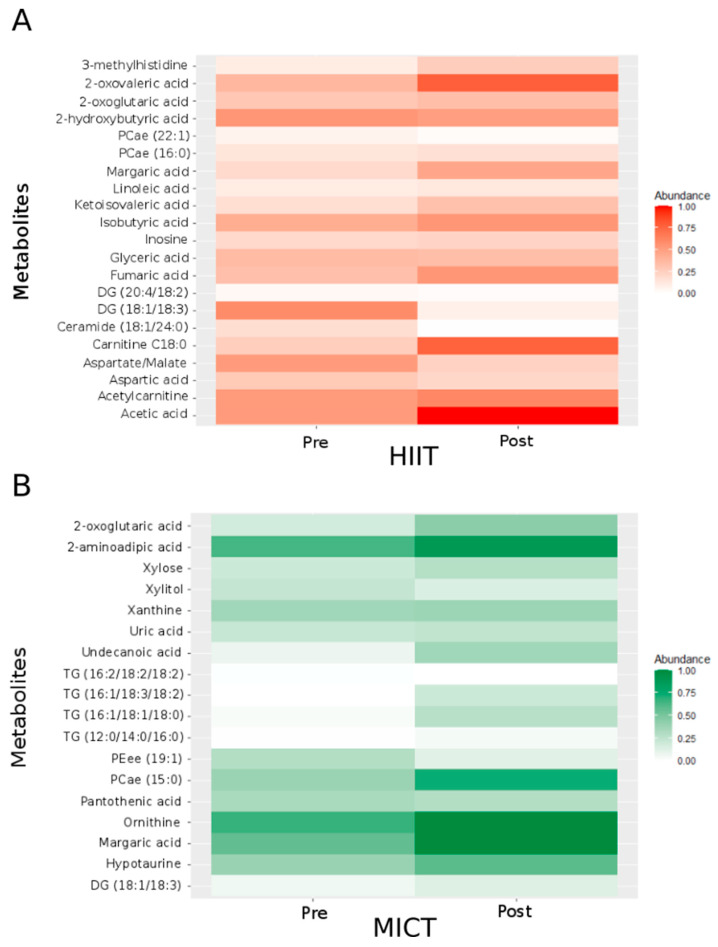
Heatmap showing the normalized abundance of the metabolites and ratios of metabolites with significant HIIT effect (**A**) and MICT effect (**B**). For normalization, each metabolite’s minimal value was subtracted from all values and divided by the highest value. Pre: before the 12-week intervention. Post: after the 12-week intervention. PCae: acyl-alkyl-phosphatidylcholine; DG: Diglyceride; TG: Triglyceride; PEee: ether-phosphatidylethanolamine; PCae: acyl-alkyl-phosphatidylcholine; / = ratio.

**Figure 3 metabolites-13-00198-f003:**
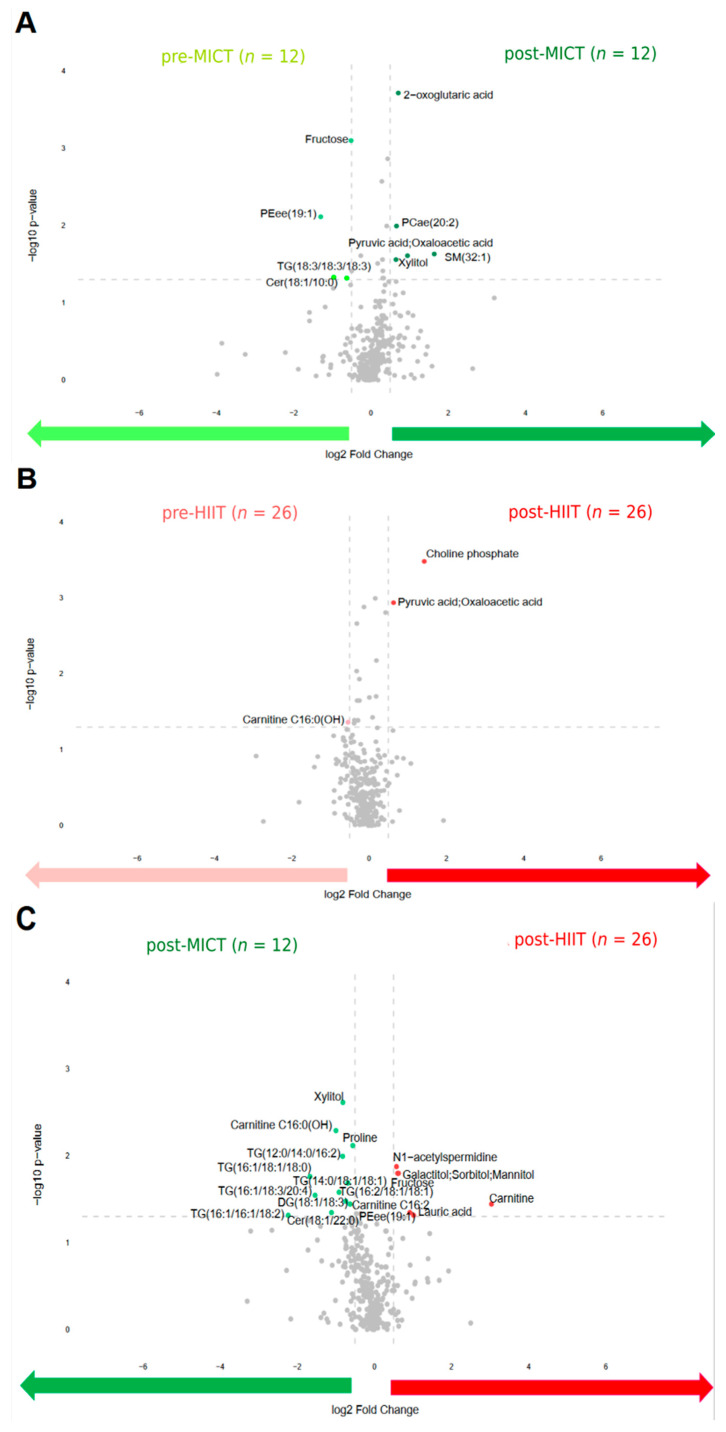
Volcano plots of all metabolites alterations following the 12-week intervention, showing the relationship between fold change (log2 Fold Change; horizontal axis) and significance (−log10 *p*-value; vertical axis). Fold changes describe alterations comparing: pre to post MICT (**A**); pre to post HIIT (**B**); post MICT to post HIIT (**C**). Pre: before the 12-week intervention; Post: after the 12-week intervention; MICT: moderate-intensity continuous training; HIIT: high-intensity interval training. Metabolites shown as light green and dark green points refer to the metabolites upregulated abundance before MICT and after MICT respectively. Metabolites shown as light red and dark red points refer to the metabolites upregulated abundance before HIIT and after HIIT respectively.

**Figure 4 metabolites-13-00198-f004:**
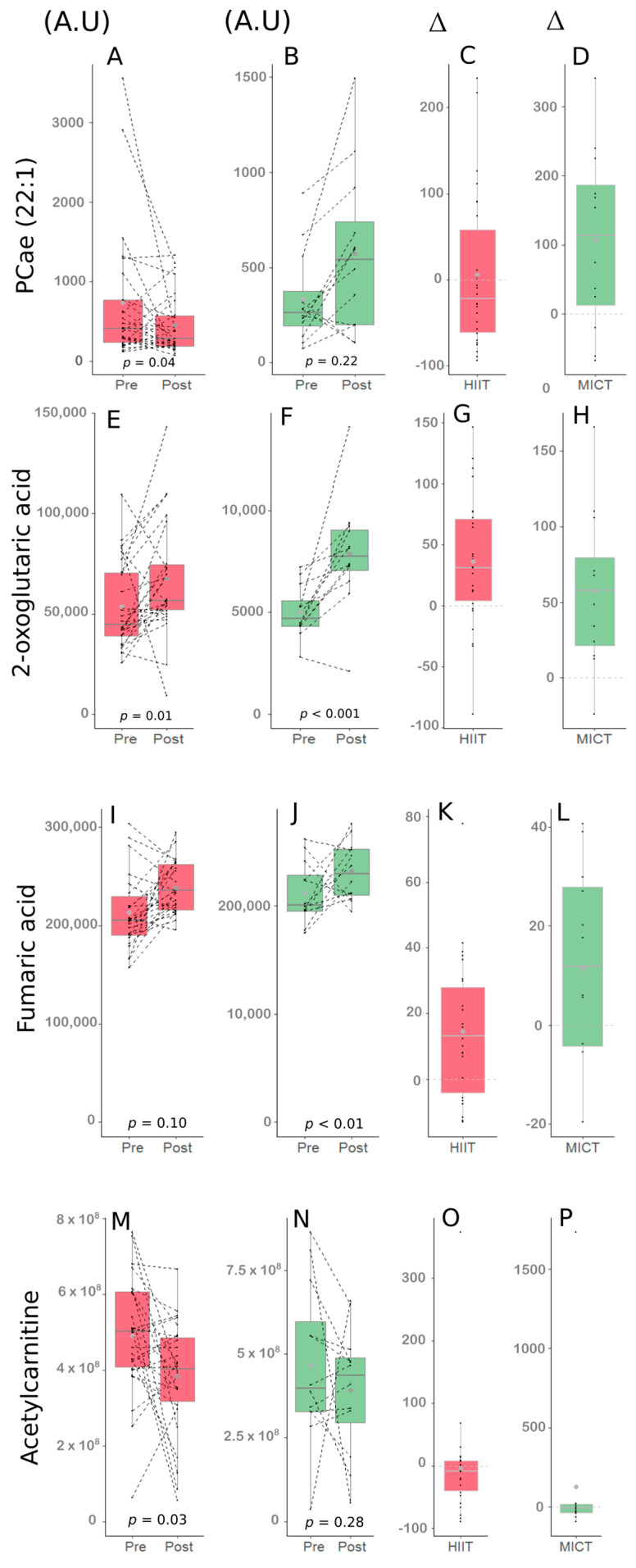
Impact of HIIT and MICT on the four metabolites of interest. HIIT: high-intensity interval training (red), MICT: moderate-intensity continuous training (green), Pre: before the 12-week intervention, Post: after the 12-week intervention, A.U: abundance unit, ∆: delta change between Pre and Post (expressed in %). PCae (22:1) following HIIT (**A**), PCae (22:1) following MICT (**B**), delta change of PCae (22:1) following HIIT (**C**), delta change of PCae (22:1) following MICT (**D**), 2-oxoglutaric acid following HIIT (**E**), 2-oxoglutaric acid following MICT (**F**), delta change of 2-oxoglutaric acid following HIIT (**G**), delta change of 2-oxoglutaric acid following MICT (**H**), Fumaric acid following HIIT (**I**), Fumaric acid following MICT (**J**), delta change of Fumaric acid following HIIT (**K**), delta change of Fumaric acid following MICT (**L**)**,** Acetylcarnitine following HIIT (**M**), Acetylcarnitine following MICT (**N**), delta change of Acetylcarnitine following HIIT (**O**), delta change of Acetylcarnitine following MICT (**P**).

**Figure 5 metabolites-13-00198-f005:**
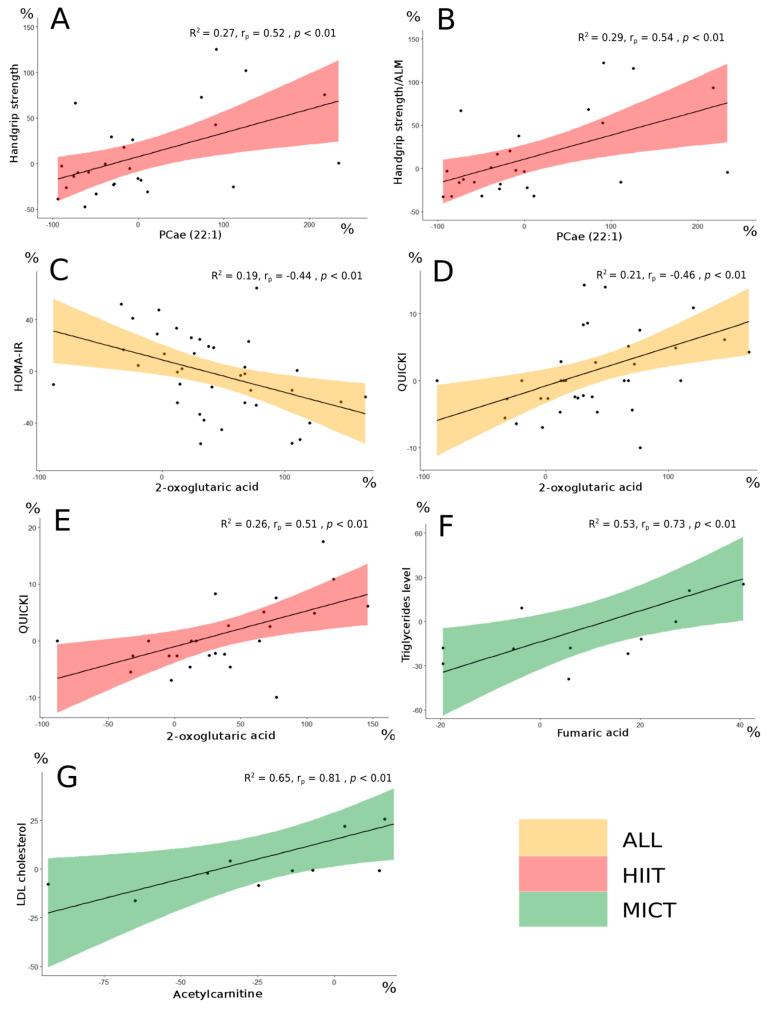
Significant correlations in the different groups between metabolites delta change (%) and clinico-biological parameters delta change (%). R^2^ is the coefficient of determination for the linear approximation. Handgrip strength and PCae (22:1) (**A**). Handgrip strength/ALM and PCae (22:1) (**B**). HOMA-IR and 2-oxoglutaric acid (**C**). QUICKI and 2-oxoglutaric acid (**D**,**E**). Triglycerides level and fumaric acid (**F**). LDL cholesterol and acetylcarnitine (**G**). PCae: acyl-alkyl-phosphatidylcholine; ALM: arms lean mass; HOMA-IR: homeostatic model assessment for insulin resistance; QUICKI: quantitative insulin-sensitivity check index; LDL: low-density lipoprotein; All: no group distinction (orange filling). HIIT: high-intensity interval training (red filling). MICT: moderate-intensity continuous training (green filling). Correlation analysis for both All and HIIT groups was performed using parametric Pearson’s test. Correlation analysis for the MICT group was performed using non-parametric Spearman’s test.; r_p_: Pearson’s correlation coefficient; r_s_: Spearman’s correlation coefficient.

**Table 1 metabolites-13-00198-t001:** Characteristics of participants at baseline.

	HIIT	MICT	*p*-Value
Age (years)	67.73 ± 3.87	70.41 ± 3.89	0.71
Men/women (*n*)	13/13	8/4	N/A
Fat mass (%—Men/Women)	33.03 ± 5.01	43.30 ± 6.38	0.64/0.60
BMI (kg/m^2^)	29.77 ± 2.96	30.08 ± 7.66	0.53
Total fat mass (%)	37.75 ± 7.93	37.34 ± 7.22	0.87
Android fat mass (%)	47.07 ± 7.46	47.47 ± 5.90	0.87
Gynoid fat mass (%)	39.88 ± 10.48	38.75 ± 11.05	0.75
Steps per day (*n*)	6658 ± 3299	6595 ± 2454	0.81
Energy intake (kcal/day)	2308 ± 466	2153 ± 521	0.39
MoCA (/30)	27.24 ± 2.23	27.76 ± 2.34	0.07

Data are presented as: mean ± SD; HIIT: high-intensity interval training; MICT: moderate-intensity continuous training; BMI: body mass index; MoCA: Montreal Cognitive Assessment.

**Table 2 metabolites-13-00198-t002:** Effect of a 12-week High-Intensity Interval Training (HIIT) and Moderate-Intensity Continuous Training (MICT) on metabolites of tricarboxylic acid (TCA) cycle, carbohydrate metabolism and amino acid metabolism in obese older adults.

Metabolites	HIIT (*n* = 26)		MICT (*n* = 12)		*p*-Value	
	Pre	Post	Pre	Post	Time Effect	Time × Group Effect
TCA cycle						
2-oxoglutaric acid	5.4 × 10^4^ ± 2.2 × 10^4^	6.7 × 10^4^ ± 2.9 × 10^4 #^	5.0 × 10^4^ ± 1.3 × 10^4^	7.9 × 10^4^ ± 2.7 × 10^4 ###^	0.0002	0.13
3-methylhistidine	1.4 × 10^4^ ± 3.9 × 10^3^	1.2 × 10^4^ ± 3.5 × 10^3 #^	1.4 × 10^4^ ± 2.3 × 10^3^	1.1 × 10^4^ ± 2.0 × 10^3^	0.006	0.79
Aspartic acid	3.7 × 10^5^ ± 2.2 × 10^5^	2.6 × 10^5^ ± 8.2 × 10^4 ##^	1.5 × 10^5^ ± 4.8 × 10^4^	2.4 × 10^5^ ± 6.6 × 10^4^	0.12	0.005
Fumaric acid	2.1 × 10^5^ ± 3.7 × 10^4^	2.4 × 10^5^ ± 2.6 × 10^4 ##^	2.1 × 10^5^ ± 2.8 × 10^4^	2.3 × 10^5^ ± 2.7 × 10^4^	0.001	0.71
Pyruvic acid and Oxaloacetic acid	3.6 × 10^5^ ± 3.0 × 10^5^	4.9 × 10^5^ ± 3.2 × 10^5^	3.6 × 10^5^ ± 1.9 × 10^5^	5.8 × 10^5^ ± 5.6 × 10^5^	0.03	0.56
Aspartate/Malate	3.82 ± 1.81	2.81 ± 1.08 ^##^	1.72 ± 0.56	2.58 ± 0.98	0.13	0.003
Carbohydrate metabolism						
Acetic acid	3.7 × 10^4^ ± 3.4 × 10^3^	3.4 × 10^4^ ± 5.3 × 10^3 #^	3.7 × 10^4^ ± 6.0 × 10^3^	3.6 × 10^4^ ± 5.3 × 10^3^	0.04	0.32
Glyceric acid	2.1 × 10^6^ ± 6.4 × 10^5^	1.6 × 10^6^ ± 3.7 × 10^5 ###^	1.6 × 10^6^ ± 5.4 × 10^5^	1.4 × 10^6^ ± 2.4 × 10^5^	<0.0001	0.09
Ribitol	1.4 × 10^5^ ± 3.0 × 10^4^	1.4 × 10^5^ ± 2.6 × 10^4^	1.3 × 10^5^ ± 2.9 × 10^4^	1.4 × 10^5^ ± 2.4 × 10^4^	0.75	0.04
Xylitol	2.8 × 10^4^ ± 8.0 × 10^3^	2.9 × 10^4^ ± 5.4 × 10^3^	2.9 × 10^4^ ± 7.9 × 10^3^	5.2 × 10^4^ ± 3.8 × 10^4 ###^	0.02	0.004
Xylose	4.1 × 10^4^ ± 1.6 × 10^4^	3.9 × 10^4^ ± 8.2 × 10^3^	3.2 × 10^4^ ± 4.7 × 10^3^	4.3 × 10^4^ ± 8.2 × 10^3 ##^	0.31	0.009
Amino acid metabolism						
2-aminoadipic acid	5.1 × 10^5^ ± 1.2 × 10^5^	4.1 × 10^5^ ± 1.6 × 10^4^	1.7 × 10^5^ ± 8.2 × 10^4^	4.7 × 10^5^ ± 1.8 × 10^5 ##^	0.03	0.08
2-hydroxybutyric acid	3.0 × 10^4^ ± 1.4 × 10^4^	3.0 × 10^4^ ± 1.4 × 10^4 #^	4.9 × 10^4^ ± 1.7 × 10^4^	3.0 × 10^4^ ± 1.4 × 10^4^	0.12	0.04
2-oxovaleric acid	6.4 × 10^3^ ± 1.4 × 10^3^	5.4 × 10^3^ ± 1.6 × 10^3 #^	6.1 × 10^3^ ± 1.1 × 10^3^	5.5 × 10^3^ ± 1.4 × 10^3^	0.01	0.57
3-hydroxybutyric acid	3.8 × 10^4^ ± 2.5 × 10^4^	2.8 × 10^4^ ± 2.0 × 10^4^	5.9 × 10^4^ ± 5.6 × 10^4^	3.8 × 10^4^ ± 2.2 × 10^4^	0.03	0.44
Ketoisovaleric acid	1.0 × 10^5^ ± 3.4 × 10^4^	1.1 × 10^5^ ± 3.0 × 10^4 ##^	8.9 × 10^4^ ± 1.8 × 10^4^	1.2 × 10^5^ ± 2.7 × 10^4^	0.002	0.05
Hypotaurine	2.1 × 10^5^ ± 8.7 × 10^4^	2.2 × 10^5^ ± 8.8 × 10^4^	1.9 × 10^5^ ± 6.6 × 10^4^	2.7 × 10^5^ ± 7.5 × 10^4 ##^	0.07	0.04
Inosine	1.9 × 10^4^ ± 6.3 × 10^3^	1.5 × 10^4^ ± 3.9 × 10^3 ###^	1.4 × 10^4^ ± 3.7 × 10^3^	1.5 × 10^4^ ± 3.5 × 10^3^	0.01	0.01
Ornithine	7.1 × 10^6^ ± 1.7 × 10^6^	7.8 × 10^6^ ± 2.4 × 10^6^	6.7 × 10^6^ ± 1.8 × 10^6^	8.5 × 10^6^ ± 2.5 × 10^6 #^	0.01	0.21
Proline	2.6 × 10^6^ ± 2.1 × 10^6^	2.3 × 10^6^ ± 1.2 × 10^6^	2.6 × 10^6^ ± 1.5 × 10^6^	3.5 × 10^6^ ± 1.2 × 10^6^	0.68	0.04
Uric acid	2.9 × 10^6^ ± 2.2 × 10^6^	2.6 × 10^6^ ± 1.4 × 10^6^	2.6 × 10^6^ ± 1.2 × 10^6^	4.1 × 10^6^ ± 2.2 × 10^6 ##^	0.36	0.002
Xanthine	3.0 × 10^4^ ± 1.4 × 10^4^	2.9 × 10^4^ ± 1.1 × 10^4^	1.9 × 10^4^ ± 6.3 × 10^3^	2.8 × 10^4^ ± 8.3 × 10^3 ##^	0.27	0.01

Data are presented as: mean of abundance unit ± SD; HIIT: high-intensity interval training; MICT: moderate-intensity continuous training; Pre: before the 12-week intervention; Post: after the 12-week intervention; TCA: Tricarboxylic Acid; /: ratio; Time and Time × Group effects are analyzed using two-way repeated measures ANOVA. ^#^: *p* < 0.05; ^##^: *p* < 0.01; ^###^: *p* < 0.001: Significant intra-group differences between pre and post intervention using two-ways repeated measures ANOVA followed by post-hoc analyses done with simultaneous tests for general linear hypotheses.

**Table 3 metabolites-13-00198-t003:** Effect of a 12-week High-Intensity Interval Training (HIIT) and Moderate-Intensity Continuous Training (MICT) on metabolites of fat metabolism in obese older adults.

Metabolites	HIIT (*n* = 26)		MICT (*n* = 12)		*p*-Value	
	Pre	Post	Pre	Post	Time Effect	Time × Group Effect
Fat metabolism						
Acetylcarnitine	4.9 × 10^8^ ± 1.6 × 10^8^	3.8 × 10^8^ ± 1.5 × 10^8 #^	4.7 × 10^8^ ± 2.4 × 10^8^	3.9 × 10^8^ ± 1.9 × 10^8^	0.02	0.71
Arachidonic acid	2.4 × 10^5^ ± 1.0 × 10^5^	2.8 × 10^5^ ± 8.6 × 10^4^	1.7 × 10^5^ ± 6.3 × 10^4^	2.2 × 10^5^ ± 7.3 × 10^4^	0.02	0.78
Butanoic acid	2.0 × 10^3^ ± 8.4 × 10^2^	1.6 × 10^3^ ± 8.2 × 10^2^	2.3 × 10^3^ ± 7.8 × 10^2^	7.8 × 10^3^ ± 7.9 × 10^2^	0.03	0.84
Carnitine C18:0	68.78 ± 38.11	120.40 ± 101.14 ^#^	84.5 ± 57.0	94.92 ± 51.95	0.04	0.55
Ceramide (18:1/22:0)	5.9 × 10^5^ ± 4.9 × 10^5^	4.0 × 10^5^ ± 3.4 × 10^5^	4.9 × 10^5^ ± 4.9 × 10^5^	8.7 × 10^5^ ± 7.6 × 10^5^	0.94	0.02
Ceramide (18:1/24:0)	8.5 × 10^6^ ± 4.7 × 10^6^	6.2 × 10^6^ ± 3.1 × 10^6 #^	6.7 × 10^6^ ± 4.3 × 10^6^	9.1 × 10^6^ ± 4.7 × 10^6^	0.39	0.03
DG (18:1/18:3)	1.3 × 10^3^ ± 7.2 × 10^2^	9.4 × 10^2^ ± 6.4 × 10^2 #^	8.9 × 10^2^ ± 8.5 × 10^2^	1.8 × 10^3^ ± 1.1 × 10^3 ##^	0.86	0.0004
DG (20:4/18:2)	8.7 × 10^2^ ± 1.1 × 10^3^	4.4 × 10^2^ ± 3.0 × 10^2 #^	7.9 × 10^2^ ± 9.9 × 10^2^	5.2 × 10^2^ ± 4.3 × 10^2^	0.04	0.67
Isobutyric acid	1.0 × 10^4^ ± 2.2 × 10^3^	8.4 × 10^3^ ± 2.9 × 10^3 ##^	8.9 × 10^3^ ± 1.1 × 10^3^	8.6 × 10^3^ ± 1.2 × 10^3^	0.007	0.15
Linolenic acid	1.4 × 10^8^ ± 3.7 × 10^8^	1.4 × 10^7^ ± 3.7 × 10^7 #^	1.4 × 10^7^ ± 3.7 × 10^7^	1.4 × 10^7^ ± 3.7 × 10^7^	0.03	0.66
Margaric acid	3.6 × 10^5^ ± 8.3 × 10^4^	4.1 × 10^5^ ± 7.5 × 10^4 #^	3.4 × 10^5^ ± 5.0 × 10^4^	4.2 × 10^5^ ± 9.6 × 10^4 ##^	0.001	0.28
Pantothenic acid	1.5 × 10^4^ ± 6.4 × 10^3^	1.5 × 10^4^ ± 6.5 × 10^3^	1.2 × 10^4^ ± 4.5 × 10^3^	1.5 × 10^4^ ± 7.6 × 10^3 ###^	0.09	0.003
PCae (15:0)	4.8 × 10^4^ ± 3.0 × 10^4^	3.5 × 10^4^ ± 1.3 × 10^4^	3.1 × 10^4^ ± 1.4 × 10^4^	4.1 × 10^4^ ± 1.5 × 10^4 #^	0.18	0.01
PCae (16:0)	4.7 × 10^6^ ± 3.1 × 10^6^	3.5 × 10^6^ ± 1.5 × 10^6 #^	3.2 × 10^6^ ± 1.1 × 10^6^	4.1 × 10^6^ ± 1.4 × 10^6^	0.19	0.02
PCae (20:2)	1.4 × 10^4^ ± 1.3 × 10^4^	1.1 × 10^4^ ± 6.8 × 10^3^	9.7 × 10^3^ ± 5.9 × 10^3^	1.5 × 10^4^ ± 7.5 × 10^3^	0.75	0.03
PCae (22:1)	7.4 × 10^2^ ± 8.3 × 10^2^	4.5 × 10^2^ ± 3.8 × 10^2 #^	3.3 × 10^2^ ± 2.5 × 10^2^	5.7 × 10^2^ ± 4.3 × 10^2^	0.29	0.03
PCae (22:4)	1.0 × 10^3^ ± 1.2 × 10^3^	7.4 × 10^2^ ± 7.2 × 10^2^	5.2 × 10^2^ ± 6.2 × 10^2^	1.2 × 10^3^ ± 1.2 × 10^3^	0.92	0.04
PEaa (36:1)	2.9 × 10^5^ ± 2.1 × 10^5^	2.2 × 10^5^ ± 1.6 × 10^5^	2.1 × 10^5^ ± 1.3 × 10^5^	3.3 × 10^5^ ± 2.3 × 10^5^	0.67	0.04
PEaa (38:6)	1.4 × 10^4^ ± 2.0 × 10^4^	7.8 × 10^5^ ± 8.3 × 10^5^	1.0 × 10^5^ ± 1.0 × 10^5^	2.1 × 10^5^ ± 2.0 × 10^5^	0.74	0.02
PEee (19:1)	2.0 × 10^2^ ± 1.6 × 10^2^	1.7 × 10^2^ ± 1.0 × 10^2^	2.6 × 10^2^ ± 2.2 × 10^2^	8.9 × 10^1^ ± 3.0 × 10^1 ##^	0.02	0.03
TG (12:0/12:0/16:1)	3.3 × 10^3^ ± 4.1 × 10^3^	1.7 × 10^3^ ± 1.5 × 10^3^	4.8 × 10^3^ ± 8.7 × 10^3^	1.7 × 10^3^ ± 1.2 × 10^3^	0.04	0.48
TG (12:0/14:0/16:0)	4.7 × 10^4^ ± 6.6 × 10^4^	2.6 × 10^4^ ± 1.8 × 10^4^	8.1 × 10^4^ ± 1.2 × 10^5^	2.5 × 10^4^ ± 2.7 × 10^4 #^	0.02	0.24
TG (14:0/16:0/16:0)	6.9 × 10^5^ ± 3.5 × 10^5^	5.5 × 10^5^ ± 2.2 × 10^5^	7.4 × 10^5^ ± 8.7 × 10^5^	5.4 × 10^5^ ± 3.7 × 10^5^	0.04	0.74
TG (14:0/16:2/16:2)	9.3 × 10^3^ ± 6.0 × 10^3^	6.8 × 10^3^ ± 5.6 × 10^3^	9.9 × 10^3^ ± 1.0 × 10^4^	6.8 × 10^3^ ± 4.6 × 10^3^	0.03	0.81
TG (16:1/18:1/18:0)	2.2 × 10^4^ ± 1.9 × 10^4^	1.5 × 10^4^ ± 1.7 × 10^4^	2.6 × 10^4^ ± 2.4 × 10^4^	5.7 × 10^4^ ± 5.5 × 10^4 ##^	0.35	0.003
TG (16:1/18:3/18:2)	5.7 × 10^5^ ± 8.7 × 10^6^	4.4 × 10^5^ ± 8.7 × 10^6^	1.5 × 10^6^ ± 8.7 × 10^6^	4.8 × 10^6^ ± 8.7 × 10^6 ###^	0.09	0.004
TG (16:1/18:3/20:4)	3.2 × 10^4^ ± 5.4 × 10^4^	1.7 × 10^4^ ± 1.4 × 10^4^	1.7 × 10^4^ ± 1.2 × 10^4^	4.8 × 10^4^ ± 5.4 × 10^4^	0.96	0.02
TG (16:2/18:2/18:2)	8.4 × 10^4^ ± 2.5 × 10^5^	7.6 × 10^4^ ± 2.2 × 10^5^	5.0 × 10^5^ ± 1.6 × 10^6^	2.2 × 10^6^ ± 3.8 × 10^6 ###^	0.04	0.002
Undecanoic acid	1.4 × 10^5^ ± 5.2 × 10^4^	9.4 × 10^4^ ± 3.4 × 10^4^	9.3 × 10^4^ ± 3.4 × 10^4^	1.2 × 10^5^ ± 4.4 × 10^4 ###^	0.06	0.001

Data are presented as: mean of abundance unit ± SD; HIIT: high-intensity interval training; MICT: moderate-intensity continuous training; Pre: before the 12-week intervention; Post: after the 12-week intervention; TCA: Tricarboxylic Acid; DG: Diglyceride; PCae: acyl-alkyl-phosphatidylcholine; PEaa: alkyl acyl-phosphatidylethanolamine; PEee: ether-phosphatidylethanolamine; TG: Triglyceride; Time and Time × Group effects are analyzed using two-way repeated measures ANOVA. ^#^: *p* < 0.05; ^##^: *p* < 0.01; ^###^: *p* < 0.001: Significant intra-group differences between pre and post intervention using two-ways repeated measures ANOVA followed by post-hoc analyses done with simultaneous tests for general linear hypotheses.

**Table 4 metabolites-13-00198-t004:** Significant correlations between functional capacities as well as muscle strength parameters delta changes and significant metabolites delta changes.

Parameters	Intervention (*n* = 38)	HIIT (*n* = 26)	MICT (*n* = 12)
6 min walking test	PEee (19:1): 0.6 ***	-	-
4 m walk test normal	TG (12:0/12:0/16:1): 0.45 **	-	-
4 m walk test fast	Ceramide (18:1/24:0): 0.49 ** DG (18:1/18:3): 0.55 *** TG (16:1/18:1/18:0): 0.61 *** TG (16:1/18:3/18:2): 0.55 *** TG (16:1/18:3/20:4): 0.61 ***	-	-
Unipodal Balance Test	-	-	Xanthine: 0.75 **
Chair test	Insosine: 0.48 ** TG (16:1/18:1/18:0): 0.44 ** TG (16:1/18:3/18:2): 0.48 **	-	3-methylhistidine: −0.73 ** TG (14:0/16:2/16:2):−0.78 **
Timed Up and Go Test	Ceramide (18:1/22:0): −0.49 ** PEaa (36:1): −0.64 ***		-
Handgrip strength	-	PCae (22:1): 0.52 **	-
Handgrip strength/BW	-	PCae (22:1): 0.54 **	-
Handgrip strength/ALM	-	PCae (22:1): 0.54 **	Ceramide (18:1/24:0): 0.76 **
Quadriceps strength/BW	-	-	Glyceric acid: −0.82 **
Quadriceps strength/LLM	-	-	2-hydroxyglutaric acid: 0.77 **
Lower Limb power	Acetic acid: 0.45 ** PCae (16:0): 0.5 **	-	-

Delta changes (%) are estimated as: [(post − pre)/pre] × 100; HIIT: high-intensity interval training, MICT: moderate-intensity continuous training; min: minutes; m: meters; BW = body weight; ALM: arms lean mass; LLM: legs lean mass; DG: diglyceride; PCae: acyl-alkyl-phosphatidylcholine; PEaa: alkyl acyl-phosphatidylethanolamine; PEee = ether-phosphatidylethanolamine; TG = triglyceride; / = ratio; Relative data are expressed in % at pre and post; Correlation analysis for both All and HIIT groups was performed using parametric Pearson’s test. Correlation analysis for the MICT group was performed using non-parametric Spearman’s test; Significance: ** *p* < 0.01; *** *p* < 0.001; -: no significant correlations.

**Table 5 metabolites-13-00198-t005:** Significant correlations between body composition parameters delta changes and significant metabolites delta changes.

Parameters	Intervention (*n* = 38)	HIIT (*n* = 26)	MICT (*n* = 12)
Body weight	Ceramide (18:1/24:0): 0.45 ** PEaa (36:1): 0.44 ** TG (16:1/18:1/18:0): 0.48 **	-	-
BMI	Ceramide (18:1/24:0): 0.45 ** PEaa (36:1): 0.44 ** TG (16:1/18:1/18:0): 0.48 **	-	-
Relative total fat mass	2-aminoadipic acid: −0.49 ** PCae (20:2): −0.5 **	Aspartic acid: −0.53 ** Aspartate/Malate: −0.54 **	Aspartic acid: 0.74 **
Relative arms fat mass	Panthotenic acid: 0.51 *** TG (16:1/18:3/18:2): 0.49 **	-	-
Relative leg fat mass	2-aminoadipic acid: −0.43 ** PCae (20:2): −0.47 **	-	TG (14:0/16:2/16:2): 0.75 **
Relative android fat mass		Acetic acid: −0.56 **	Proline: 0.75 ** Ribitol: 0.72 **
Relative gynoid fat mass	2-aminoadipic acid: 0.52 *** Acetylcarnitine: 0.42 ** Aspartic acid: −0.44 ** PCae (20:2): −0.46 **	Aspartic acid: −0.54 ** Aspartate/Malate: −0.58 **	PEee (19:1): 0.78 ***
Total lean mass	-	-	Fumaric acid: 0.86 **

Delta changes (%) are estimated as: [(post − pre)/pre] × 100; HIIT: high-intensity interval training, MICT: moderate-intensity continuous training; BMI: body mass index; PCae: acyl-alkyl-phosphatidylcholine; PEaa: alkyl acyl-phosphatidylethanolamine; PEee = ether-phosphatidylethanolamine; TG = triglyceride; /: ratio. Relative data are expressed in % at pre and post; Correlation analysis for both All and HIIT groups was performed using parametric Pearson’s test; Correlation analysis for the MICT group was performed using non-parametric Spearman’s test. Significance: ** *p* < 0.01; *** *p* < 0.001; -: no significant correlations.

**Table 6 metabolites-13-00198-t006:** Significant correlations between biological parameters delta changes and significant metabolites delta changes.

Parameters	Intervention (*n* = 38)	HIIT (*n* = 26)	MICT (*n* = 12)
Adiponectin/Adipoleptin	Hypotaurine: 0.44 **	-	-
Adiponectin	-	-	TG (16:1/18:3/20:4): 0.78 **
Leptin	-	-	3-hydroxubutyric acid: −0.78 **
Free fatty acids	Acetic acid: 0.45 ** Ketoisovaleric acid: 0.43 ** Linoleic acid: 0.6 ***	3-methylhistidine: 0.61 **	-
Total cholesterol	TG (14:0/16:2/16:2): 0.46 **	-	PEaa (38:6): −0.79 **
LDL cholesterol	-	-	Acetylcarnitine: 0.81 **
Triglycerides	Carnitine C18:0: 0.71 **	-	Fumaric acid: 0.73 **
Ferritin	-	-	Uric acid: −0.77 **
IGF1	-	-	Ornithine: −0.78 **
IGF1/IGFBP3	Butanoic acid: −0.44 ** Aspartate/Malate: 0.46 **	-	TG (16:1/18:1/18:0): −0.72 **
Glucose	PCae (22:4): −0.46 **	Linoleic acid: −0.54 **	-
HOMA-IR	2-oxoglutaric acid: −0.44 **	Linoleic acid: −0.52 **	-
QUICKI	2-oxoglutaric acid: 0.46 **	2-oxoglutaric acid: 0.51 **	-

Delta changes (%) are estimated as: [(post − pre)/pre] ×100; HIIT: high-intensity interval training, MICT: moderate-intensity continuous training; Intervention: both groups together; HOMA-IR: homeostatic model assessment for insulin resistance; LDL: low-density lipoprotein; IGF1: insulin-like growth Factor-1; IGFBP3: insulin-like growth factor binding protein-3; QUICKI: quantitative insulin-sensitivity check index; PCae: acyl-alkyl-phosphatidylcholine; TG: triglyceride; /: ratio; Relative data are expressed in % at pre and post. Correlation analysis for both All and HIIT groups was performed using parametric Pearson’s test; Correlation analysis for the MICT group was performed using non-parametric Spearman’s test; Significance: ** *p* < 0.01; *** *p* < 0.001; -: no significant correlations.

## Data Availability

The datasets used during the current study are available from the corresponding author on reasonable request. The data are not publicly available due to [the fact that data required to reproduce the above results are also part of an ongoing study.

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
