# Peer review of "Serum Metabolome Adaptations Following 12 Weeks of High-Intensity Interval Training or Moderate-Intensity Continuous Training in Obese Older Adults"

_metabolites, 2023, doi:10.3390/metabo13020198_

Round 1

Reviewer 1 Report

Quick summary of the article

The manuscript “Serum metabolomic adaptations following 12 weeks of High-Intensity 2 Interval Training or Moderate-Intensity Continuous Training in obese 3 older adults” proposed a study of serum metabolome analysis with mass spectroscopy coupled with gas or liquid chromatography. The main outcome was the effect of two type of training during 12 weeks on obese older adults. The authors highlighted 51 metabolites; 21 metabolites changes after HIIT and 18 after MICT and porposed correlation with biological clinical parameter.

General comments

Originality and scientific quality

It is a rare study on human serum with LC or GC-MS on older obese adults. Moreover, it is related to clinical and biological parameters. It is well performed but we regret a lack of detail in the materials and methods section in the main text (detailed procedures allowing exact reproduction of spectroscopy experiment and data processing).

Pertinence for the journal

The document is totally in accordance with subjects treated in Metabolites.

Presentation and structure

The structure in this article is respected. Figures are well presented but the resolution of the text in the figure is poor.

The authors would check double spaces.

Title and keywords

We suggest also to add LC-MS GC-MS/MS or mass spectroscopy in the keywords.

Materials and methods

Section 2.5 : Police character problem.

As said earlier, detailed procedures allowing exact reproduction of experiment are needed (for biological parameter and metabolomic profiling) even if previously described in the reference 26 : column, workflow … Same thing for multivariate statistical analysis : data processing, bucketing, standardization, normalization …

Results

Line 278 to 288 : Section 2.5 : Police character problem.

There is a lot of information in the tables and it is difficult to interpret; is it possible to propose a color code or to sort metabolites to directly see those who increase from those who decrease?

Are the PCA really informative or useful to determine which metabolites were discriminant as they revealed no significant change? Maybe this figure is unnecessary in the main text.

Correlation between metabolite and biological parameter are very interesting. However, table 7 is overloaded. Is it possible to separate data in two or three table (Intervention table 7a; HIIT and MICT table 7b and eventually 7c)

Discussion

Discussion is well conduced.

Recommendation

Minor revision

Author Response

We thank the reviewer for his comments and suggestions which were very helpful.
Please see the attachment.

Reviewer 2 Report

The manuscript submitted by Youssef et al. uses metabolomics to unravel changes in serum metabolites in two cohorts of obese, older adults subjected to 12 weeks of high intensity interval training (HIIT) or moderate intensity continuous training (MICT). The authors provided a good overall study design and clearly address limitations of the study, as well as suggestions for future studies to increase statistical power and improve the study design.

A few minor comments for the authors to address:

Tables:

-       Overall, I think there are too many tables in the main paper. Tables 2-4 should be placed in the supplementary portion of the paper, as these tables focus on the clinical/anthropometric characteristics of the subjects rather than the metabolomic changes – which should be the focus of the paper.

Figure 1: PCA Plots

-       Showing 4 different PCA plots is a bit redundant and confusing. I would show one PCA plot with all four groups: i) Pre-HIIT, ii) Post-HIIT, iii) Pre-MICT, iv) Post-MICT.

Figure 2: Heatmaps

-       I’m not sure this visualization is the best way to illustrate changes in metabolites pre vs. post changes. Some metabolites illustrated in the heatmap based on the color scheme showed no changes. For example, in Figure 2A, when comparing changes in Pre and Post-HIT, some metabolites showed almost no changes based on normalized abundances such as: PCae (22:1) and DG (20:4/18:2). Furthermore, some of these trends illustrated in the heatmaps contradict what is stated in the text. For instance, the authors stated xylitol and xylose significantly increased after MICT, however based on the color scheme in the figure, the changes were subtle and in the case of xylitol, post-changes seem to have decreased based on the color scheme.

-       The authors need to clearly state how the abundances were normalized. What transformations/scaling was used?

-       I would recommend the authors use a different visualization tactic to illustrate trends in the data. The Volcano Plot in Figure 3 is a good way to show key metabolites. Otherwise, bar/boxplots of a few metabolites would suffice.

Discussion:

-       The authors stated that one of the main objectives of this manuscript was to identify new putative biomarkers specific to HIIT and MICT in older, obese adults. However, after thoroughly reading the discussion, it seems like the authors emphasized more on correlating the metabolomics data to clinical parameters such as muscle strength and/or glycemic parameters. From a metabolism point of view, I think it would be insightful for the authors to comment on how changes in specific metabolites highlighted in the text, specifically in pathways such as TCA cycle, lipid and carbohydrate metabolism, change due to HIIT and MICT.

Overall, the manuscript was concise and adequately written. The authors should focus on revising the tables and figures in the paper to accurately depict what is written in the text as well as elaborate on metabolic changes seen in both exercise regimes in the context of exercise metabolism.

Author Response

We thank the reviewer for his comments and suggestions which were very helpful.
Please see the attachment

Reviewer 3 Report

Between 2015 and 2050, the percentage of people over 60 will rise from 12% to 22% worldwide.

In low- and middle-income countries, according to the World Health Organization, 80% of the old population will live by 2050.

This is a crucial group because of the importance of the older population and the high cost of healthcare.

This research is essential because it focuses on a detailed analysis of metabolic parameters in the elderly during HIIT and MICT and depends on the favorable effects of exercise as a readily available and affordable method on this demographic.

All of the grammar and abstract are simple and understandable.

Exact and clear tables provide a comprehensive explanation of the methods and results.

The discussion is organized and comprehensive.

 It is only recommended that following articles should be added to the discussion.

·       Schranner D, Schönfelder M, Römisch‐Margl W, Scherr J, Schlegel J, Zelger O, Riermeier A, Kaps S, Prehn C, Adamski J, Söhnlein Q. Physiological extremes of the human blood metabolome: A metabolomics analysis of highly glycolytic, oxidative, and anabolic athletes. Physiological Reports. 2021 Jun;9(12):e14885.

·       Themistocleous IC, Agathangelou P, Stefanakis M. Retrospective Comparison of Two Circuit Training Programs with Different Intensities in Obese and Overweight Individuals. Int J Sports Exerc Med. 2023;8:240.

Author Response

We thank the reviewer for his suggestion which was very helpful.
Please see the attachment.
